# Durability of Antibody Response after Primary Pneumococcal Double-Dose Prime-Boost Vaccination in Adult Kidney Transplant Recipients and Candidates: 18-Month Follow-Up in a Non-Blinded, Randomised Clinical Trial

**DOI:** 10.3390/vaccines10071091

**Published:** 2022-07-07

**Authors:** Lykke Larsen, Claus Bistrup, Søren Schwartz Sørensen, Lene Boesby, Charlotte Sværke Jørgensen, Christian Nielsen, Isik Somuncu Johansen

**Affiliations:** 1Department of Infectious Diseases, Odense University Hospital, 5000 Odense, Denmark; isik.somuncu.johansen@rsyd.dk; 2Department of Clinical Research, University of Southern Denmark, 5230 Odense, Denmark; 3OPEN, Open Patient Data Explorative Network, Odense University Hospital, 5000 Odense, Denmark; 4Department of Nephrology, Odense University Hospital, 5000 Odense, Denmark; claus.bistrup@rsyd.dk; 5Clinical Institute, University of Southern Denmark, 5230 Odense, Denmark; 6Department of Nephrology, Rigshospitalet, Copenhagen University Hospital, 2100 Copenhagen, Denmark; soeren.schwartz.soerensen@regionh.dk; 7Department of Clinical Medicine, University of Copenhagen, 2200 Copenhagen, Denmark; 8Department of Medicine, Zealand University Hospital Roskilde, 4000 Roskilde, Denmark; lboes@regionsjaelland.dk; 9Department of Virus and Microbiological Special Diagnostics, Statens Serum Institut, 2300 Copenhagen, Denmark; csv@ssi.dk; 10Department of Clinical Immunology, Odense University Hospital, 5000 Odense, Denmark; christian.nielsen@rsyd.dk

**Keywords:** 13-valent pneumococcal conjugate vaccine, 23-valent pneumococcal polysaccharide, kidney transplant recipient, immunogenicity

## Abstract

Background: Pneumococcal prime-boost vaccination is recommended for solid organ transplant recipients and candidates. The long-term durability of the antibody (AB) response is unknown. The same applies to a dose-dependent immune response. Methods: We studied the durability of the vaccine response after 18 months in kidney transplant recipients (KTRs) and patients on the kidney transplant waiting list (WLPs). Both groups received either a normal dose (ND) or a double dose (DD) of the 13-valent pneumococcal conjugate vaccine and the 23-valent pneumococcal polysaccharide vaccine. The average pneumococcal AB geometric mean concentration (GMC) was evaluated. A level ≥ 1 mg/L was considered protective against invasive pneumococcal disease (IPD). Results: Sixty WLPs and 70 KTRs were included. The proportion of participants protected declined from 52% to 33% in WLPs and from 29% to 16% in KTRs, with the previously significant dose-effect in WLPs no longer present (40% DD vs. 27% ND; *p* = 0.273). Average pneumococcal AB GMCs remained significantly above baseline levels (all groups *p* ≤ 0.001). Drug-induced immunosuppression diminished the vaccine dose-effect. Conclusions: At follow-up, the pneumococcal prime-boost vaccination still provided significantly elevated average pneumococcal AB GMCs in both populations. Though the proportion of participants protected against IPD in WLP-DD and WLP-ND were statistically comparable, a DD may still be recommended for WLPs (EudraCT: 2016-004123-23).

## 1. Introduction

The incidence of invasive pneumococcal disease (IPD) in kidney transplant recipients (KTRs) has been estimated to be approximately nine times higher than in the general population [1]. Hence, immunisation against pneumococci is recommended for KTRs and patients eligible for a kidney transplant. Data suggest that KTRs as well as patients with end-stage kidney disease (ESKD) may have a suboptimal pneumococcal vaccine response and a rapid decline in pneumococcal antibodies (ABs) compared with healthy controls [2,3,4]. For both patient populations, this could be caused by immune dysfunction, for example, renal failure for patients with ESKD [5] or lifelong immunosuppressive treatments primarily targeting cellular immunity for KTRs [6,7]. In Denmark, the 23-valent pneumococcal polysaccharide vaccine (PPV23) is recommended for everyone over 64 years and nursing home residents or the equivalent, as well as selected populations with a particularly high or increased risk of developing IPD. People with a particularly high risk of IPD are also recommended a pneumococcal conjugate vaccine. Hence, KTRs are recommended a pneumococcal conjugate vaccine followed by PPV23 after a minimum of 8 weeks by Danish [8] and international guidelines [9,10] (until this year, the 13-valent pneumococcal conjugate vaccine (PCV13) was used, but now 15-valent and 20-valent conjugate vaccines are also available). The PPV23 may be repeated after 5–6 years. Private practitioners have been responsible for vaccinations. However, we recently showed that under 4% of KTRs were vaccinated [11]. The pneumococcal prime-boost vaccine method is based on studies including patients with sickle cell anaemia, human immunodeficiency virus (HIV), and Hodgkin’s lymphoma, where it was considered superior to PPV23 alone [12,13,14]. The PCV13 induces a T-cell response, thereby promoting B-cell differentiation into both memory B-cells and antibody-secreting plasma cells. This production of memory B-cells is thought to enhance the PPV23 response. The PPV23, consisting of purified pneumococcal polysaccharides, induces a restricted IgG response without recruiting T-cells or generating memory B-cells [15]. Immunogenicity from the pneumococcal prime-boost vaccination is not well-described in KTRs or patients with ESKD, nor is the optimal time interval between the vaccines known. Studies have demonstrated a dose effect with pneumococcal vaccines in healthy adults that enhanced immunogenicity [16,17,18]. We recently conducted a randomised trial to evaluate the eventual dose effect in patients on the kidney transplant waiting list (WLPs) and KTRs [19]. We demonstrated that twice the standard dose of both pneumococcal vaccines (PCV13 and PPV23 at 12 weeks apart) resulted in significantly more WLPs reaching the estimated protective AB level at five weeks post-vaccination compared to the standard dose. In KTRs, a dose effect was not observed. No long-term data exists on the durability of the AB response in KTRs and WLPs following a pneumococcal prime-boost vaccination either with standard or double-dose vaccines. To address this, a follow-up trial was conducted. The evaluation was carried out approximately 18 months after PPV23. In addition, we assessed any impact on the vaccine response of immunosuppression in WLPs and of baseline T-, B-, and NK cell levels.

## 2. Materials and Methods

### 2.1. Participants and Interventions

This is an 18-month follow-up study in a phase III, multi-centre, parallel-group, non-blinded, randomised, clinical trial [19]. Participants were adult KTRs and WLPs. KTRs had all received their allografts within the preceding 18 months. For WLPs, former kidney transplantation or immunosuppression were not exclusion criteria. However, the function of the graft had to be extinct. Participants were vaccinated with PCV13 followed by PPV23 after 12 weeks. They were randomised into two parallel arms: (i) normal dose (ND), 0.5 mL of both vaccines or (ii) double dose (DD), 1.0 mL of both vaccines. Blood samples were drawn at baseline and weeks 12, 17, 48, and 96. Last visit was January 2021. The primary endpoint in the original trial was immunogenicity of the pneumococcal prime-boost vaccination in two different doses 5 weeks post-PPV23 (week 17).

### 2.2. Vaccines Administered

PCV13 (Pfizer), 0.5 mL, contains polysaccharides of serotypes 1, 3, 4, 5, 6A, 6B, 7F, 9V, 14, 18C, 19A, 19F, and 23F; each one is individually conjugated to CRM197 (non-toxic mutant of diphtheria toxin). The vaccine contains 4.4 µg of 6B and 2.2 µg of each remaining saccharide. PPV23 (Sanofi Pasteur), 0.5 mL, consists of purified capsular polysaccharide. PPV23 contains 25 µg of each pneumococcal serotype: 1, 2, 3, 4, 5, 6B, 7F, 8, 9N, 9V, 10A, 11A, 12F, 14, 15B, 17F, 18C, 19A, 19F, 20, 22F, 23F, and 33F. Vaccines were administered intramuscularly.

### 2.3. Laboratory Methods

At each study visit, pneumococcal serotype-specific IgG antibody (SS IgG AB) concentrations were measured for 12 serotypes (1, 3, 4, 5, 6B, 7F, 9V, 14, 18C, 19A, 19F, and 23F) included in both pneumococcal vaccines using an in-house Luminex method [20]. This procedure enables the concurrent multiplex measurement of all 12 analytes in one sample. The samples were analysed in duplicate and re-tested if the coefficient of variation between them was above 20%. AB levels were reported as mg/L. Values above the range of the standard curve were appointed a value of 50 mg/L.

At baseline, counting of T-, B-, and NK-cells was performed using fresh EDTA blood stained with the BD Multitest™ 6-color TBNK reagent in BD TruCount tubes. Samples were analysed on a BD FACSCanto™ II flow cytometer with BD FACSDiva software.

### 2.4. Endpoints

Participants obtaining a protective response (PR) at week 96 were evaluated. A PR was defined as an average AB geometric mean concentration (GMC) ≥ 1 mg/L based on 12 vaccine-shared pneumococcal SS IgG ABs. In Denmark, this cut-off level is used when assessing whether a PPV23 vaccinated adult is protected against IPD or not [21]. Furthermore, the specific level of the average AB concentration was evaluated.

### 2.5. Statistical Analysis

SS IgG AB concentrations were logarithmically transformed for statistical purposes and reported as GMCs with 95% confidence intervals in the tables and figures. The average AB GMC was calculated as an exponentially transformed mean across the 12 logarithmically transformed SS IgG ABs. This was carried out for each participant at each visit. It was assessed as a binary variable when evaluating PR (average pneumococcal AB GMC ≥ 1mg/L), otherwise, it was assessed as a continuous variable. For the remaining continuous variables, the median with interquartile range (IQR) were reported. Student’s t-tests were used for normally distributed continuous variables. The Wilcoxon rank-sum test was used for non-normally distributed, non-paired continuous variables or the Wilcoxon signed-rank test was used if paired. Normality was judged with the Shapiro–Wilk’s test. For categorical and dichotomous variables, numbers and percentages were listed relative to patients in the groups. They were analysed using Chi-squared, McNemar’s, or Fisher’s exact test as appropriate. Spearman’s correlation was used for pairwise correlations between continuous variables. A two-sided *p*-value ≤ 0.05 was viewed as statistically significant. Statistical analyses were done using STATA 17 (Stata Corp, College Station, TX, USA).

## 3. Results

### 3.1. Trial Population

In the original trial, 142 participants were randomised. Three (2%) dropped out before receiving any vaccines. A total of 4 (2.8%) received only PCV13 and 135 (96%) received both vaccines (flow diagram for the study is depicted in Figure 1). These participants were all included when assessing for correlations between the vaccine response post-immunisation and the lymphocyte cell count. Nine participants were excluded afterwards as they were not available for the week 96 follow-up due to the following reasons: withdrawal of consent (n = 4), death (n = 4), or dropout (n = 1). Hence, at week 96, 60 WLPs (WLP-ND, n = 30; WLP-DD, n = 30) and 70 KTRs (KTR-ND, n = 37; KTR-DD, n = 33) were available for the follow-up and were included for the durability assessment of the AB response. Their median age was 52 years (IQR: 41–61) and 69.2% (90/130) were male. The baseline characteristics are depicted in Table 1. At baseline, 6 (20%) WLP-ND and 11 (36.7%) WLP-DD participants had received an immunosuppressant (*p* = 0.152). During the study, 31 (51.7%) WLPs received a kidney transplant (Table 1).

### 3.2. Protective Response

The proportion of participants with a PR is presented in Table 2 and Figure 2. At week 17, significantly more participants in WLP-DD had achieved a PR compared to WLP-ND (*p* = 0.008). However, at week 96, the groups were statistically comparable (*p* = 0.273). KTR-DD and KTR-ND were statistically comparable at every visit. The proportion of participants in WLP-DD with a PR declined significantly from week 17 to week 96 (71.4% to 40%; *p* = 0.008). In accordance with the results at week 17, significantly more WLPs had a PR at week 96 compared to KTRs (33.3% vs. 15.7%; *p* = 0.019), regardless of vaccine dose.

### 3.3. Average Pneumococcal Antibody GMC Levels

The average AB GMC levels are displayed in Table 3 and illustrated in Figure 3. All four treatment groups had an average AB GMC level significantly higher at week 17 (all groups *p* ≤ 0.001) and at week 96 (all groups *p* ≤ 0.001) compared to baseline, though both WLP groups displayed a significant decline from week 17 to week 96 (both groups *p* ≤ 0.001). Overall, regardless of vaccine dose, WLPs had significantly higher average AB GMC levels at week 17 (*p* ≤ 0.001) and week 96 (*p* = 0.019) compared to the KTRs. In Appendix A, all 12 individual SS IgG AB GMCs are depicted for each of the four treatment groups.

### 3.4. Drug-Induced Immunosuppression in WLPs

At week 17, there were 25 WLPs with drug-induced immunosuppression, either from receiving immunosuppressive medication of any kind since baseline and/or from receiving a kidney transplant between baseline and week 17 (WLP-ND = 13, WLP-DD = 12). The remaining 35 WLPs were immunocompetent except for kidney failure. Though statistically comparable, there was a tendency for immunosuppressed WLPs to have fewer participants with a PR than immunocompetent WLPs at both week 17 (41.7% vs. 61.8%; *p* = 0.131) and week 96 (28% vs. 37.1%; *p* = 0.459). Significantly more immunocompetent WLP-DD participants had a PR at week 17 compared to immunocompetent WLP-ND (*p* = 0.032), but not at week 96 (Figure 4). Immunosuppressed WLP-DD and immunosuppressed WLP-ND were comparable at both visits.

### 3.5. T, B, and NK Lymphocyte Subpopulations

In the original study, baseline absolute lymphocyte and T, B, and NK lymphocyte subpopulation counts were available in 32 WLPs and 58 KTRs. The lymphocyte cell counts are depicted in Appendix A. Compared with KTRs, WLPs had a significantly higher absolute lymphocyte cell count (median cells 1.28 × 10^3^/µL (IQR: 1.06–1.65) vs. 1.06 × 10^3^/µL (IQR:0.77–1.58); *p* = 0.028), CD4+ T lymphocytes (median cells 0.66 × 10^3^/µL (0.47–0.78) vs. 0.53 × 10^3^/µL (IQR: 0.27–0.72); *p* = 0.046), and NK cells (median cells 0.18 × 10^3^/µL (0.12–0.26) vs. 0.12 × 10^3^/µL (0.09–0.17); *p* = 0.007). In the remaining subpopulations, there were no significant differences. We found no pairwise correlations between average AB GMC levels at week 12 (PCV13) or week 17 (PCV13 + PPV23) and absolute lymphocyte cell count or any subpopulation lymphocyte cell count for either KTRs or WLPs.

## 4. Discussion

In the present study on pneumococcal prime-boost vaccinations in WLPs and KTRs including different doses of both vaccines, we found that even though WLPs benefitted short-term from a DD, the dose effect was no longer significant 18 months later when evaluating the AB response. Baseline levels of T-, B-, and NK-cells did not seem to affect the vaccine response. However, immunosuppressive medication did.

Our results are consistent with Jackson et al. who found that significant differences in ABs present 4 weeks post-vaccination between the 0.5 mL and 1.0 mL doses of the 7-valent pneumococcal conjugate vaccine were no longer present after one year [18]. An influencing factor may also be the rapid decline in pneumococcal ABs in WLPs. Previous studies have shown a significant, adequate, initial pneumococcal vaccine response in patients on dialysis, with a rapid decrease over time [4,22,23]. Kidney transplantations performed during the study may also be a contributing factor, as we observed that the difference in the PR between the DD and ND at week 17 was only significant for immunocompetent WLPs. A dose effect was not observed in KTRs at week 96, similar to week 17, as KTR-DD performed inadequately overall. In the original study, we found no clear cause for these unexpected results.

In the present study, we also showed that the average pneumococcal AB GMC level declined over time for both WLPs and KTRs. However, the week 96 levels in all 4 groups were still significantly above baseline levels. The same was true for the 12 SS IgG ABs, which were still significantly elevated above baseline values, except for four serotypes for WLP-DD (3, 9V, 19A, and 23F). This indicates a certain durability of the pneumococcal prime-boost vaccine response and is consistent with prior studies on single pneumococcal vaccines in KTRs and patients on dialysis. In KTRs, Kumar et al. [24] demonstrated that 6/7 SS IgG ABs were still significantly higher three years after the 7-valent pneumococcal conjugate vaccine compared to baseline. Oesterreich et al. found global pneumococcal ABs to be significantly above baseline levels one year after PCV13 [25], and Marrie et al. found similar levels in 11 out of 12 serotypes one year after the 14-valent pneumococcal polysaccharide vaccine [26] in KTRs. In patients on dialysis, one year after receiving either PCV13 or PPV23, Vandecasteele et al. [27] found that all SS IgG ABs were still significantly higher compared to baseline. Contrary to this, Mitra et al. demonstrated that one year after PCV13, patients on dialysis only had significantly increased AB concentration in 4 out of 13 serotypes compared to baseline [4].

Most studies reported that the pneumococcal vaccine response in patients on dialysis is better or equivalent to that in KTRs [2,28,29,30,31,32], whereas a few studies found KTRs to have a better response compared to patients on dialysis [33,34]. In the present study, the post-immunisation average pneumococcal AB GMC levels were lower in KTRs compared to WLPs throughout the entire study. This may be influenced by KTRs in our study being relatively newly transplanted and thereby heavily immunosuppressed. The timing of the vaccination may thus have been suboptimal in relation to obtaining the best vaccine response. American guidelines recommend waiting until 3 months post-transplant before vaccinating [9] but also assess that it is safe to give an influenza vaccine as early as 1 month post-transplant. However, the correct timing for pneumococcal vaccinations is not known nor is the time interval between the two vaccines when using the prime-boost vaccination approach. We do not know whether KTRs would have also exhibited a vaccine dose effect had we vaccinated them later post-transplant or had waited for a longer time between the two vaccines. The results also indicate that a booster vaccine may be required sooner than after five years in this population, which should be explored.

Compared to KTRs, WLPs seemed to experience a more rapid decline in ABs. This concurs with previous studies that showed that the decline in the pneumococcal vaccine response is faster in patients on dialysis, compared to KTRs [2,32,33,34]. Pneumococcal AB levels are the only surrogate markers for protection against IPD. For children, the World Health Organization has defined a PCV13 post-immunisation level of SS IgG ABs ≥ 0.35 mg/L to be an adequate protective response against IPD [35]. This level was derived from an analysis of conjugate vaccine efficacy data. There is no consensus on the protective levels for adult solid organ recipients or adults in general as serological assays have not been used in PPV23 efficacy studies [36]. In the diagnostics of primary immune deficiencies, a post-immunisation AB level ≥ 1.3 mg/L in 70% of the tested serotypes is viewed as protective [36,37]. In line with national standard practice, we had selected the protective AB response in adults after vaccination to be an average pneumococcal AB GMC ≥ 1.0 mg/L calculated from the 12 measured SS IgG ABs [21]. This corresponds to a minimum of 1 mg/L in each serotype but still leaves room for serotype-specific differences in immunogenicity [35,37]. Evaluating vaccine efficacy following a DD of the pneumococcal vaccines in WLPs would require a long follow-up period and a very large patient cohort. Therefore, it has never been done for KTRs or WLPs.

Cellular immunity is the main target of most immunosuppressive agents for KTRs, including cyclosporine and tacrolimus, and it has been suggested that lymphocytes, especially CD4+ T-cells, may decrease due to this treatment [38]. Accordingly, we found KTRs to have a significantly lower CD4+ T-cell count compared to WLPs. In people living with HIV, the CD4+ T-cell count has been found to be positively correlated with the AB response following T-cell-inducing vaccines [39,40,41,42] but not after the T-cell independent PPV23 [42]. In KTRs, a previous study found no correlation between the PPV23 AB response and the absolute lymphocytes’, CD3+, CD4+, or CD8+ T-cell counts [43]. In patients on haemodialysis, a higher absolute lymphocyte count was significantly associated with a positive serology after the SARS-CoV-2 (Pfizer-BioNTech) vaccine [44]. We found no correlations between the AB response after PCV13 or after both pneumococcal vaccines and any of the measured lymphocyte subpopulations in any patient group.

Our study had several limitations. Firstly, it is a follow-up study on a randomised trial where the primary endpoint was a PR 5 weeks post-PPV23. We did, however, use the same outcome at follow-up. Secondly, the primary study was underpowered, as we were not able to enrol the participants needed. Consequently, firm conclusions cannot be drawn but further studies should explore the findings made here. Thirdly, numerous WLPs underwent kidney transplantation during the study, which undoubtedly affected the follow-up results. Lastly, SS IgG ABs were used as surrogate markers for assessing vaccine efficacy, and we did not perform an opsonophagocytic assay to assess AB functionality. This could have added further knowledge when evaluating the vaccine dose-response in WLPs.

## 5. Conclusions

In KTRs and WLPs, pneumococcal prime-boost vaccinations resulted in significantly elevated average pneumococcal AB GMCs 18 months after immunisation compared to baseline. However, the number of participants estimated to be protected against IPD dropped. Although WLP-DD and WLP-ND were statistically comparable at follow-up, a DD of the pneumococcal prime-boost vaccination might still be recommendable for WLPs, as a larger proportion of WLP-DD was considered protected at follow-up and their average pneumococcal AB GMC seemed higher compared to WLP-ND.

## Figures and Tables

**Figure 1 vaccines-10-01091-f001:**
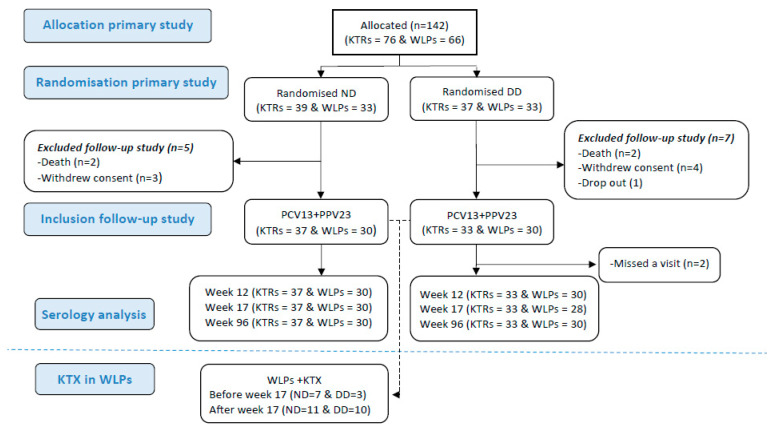
Flow diagram of patients through the study. Abbreviations: KTR, kidney transplant recipient; WLP, patients on the kidney transplant waiting list; ND, normal dose; DD, double dose; PCV13, 13-valent pneumococcal conjugate vaccine; PPV23, 23-valent pneumococcal polysaccharide; KTX, kidney transplantation.

**Figure 2 vaccines-10-01091-f002:**
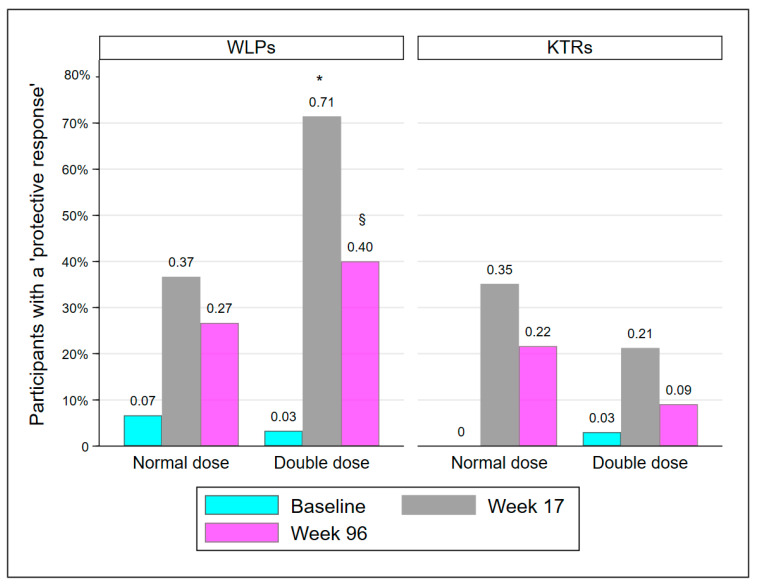
Participants protected at baseline, week 17, and week 96. Abbreviations: WLP, patients on the kidney transplant waiting list; KTR, kidney transplant recipient. *: *p* < 0.01, compared to WLP-ND week 17, §: *p* < 0.05, compared to week 17 within group.

**Figure 3 vaccines-10-01091-f003:**
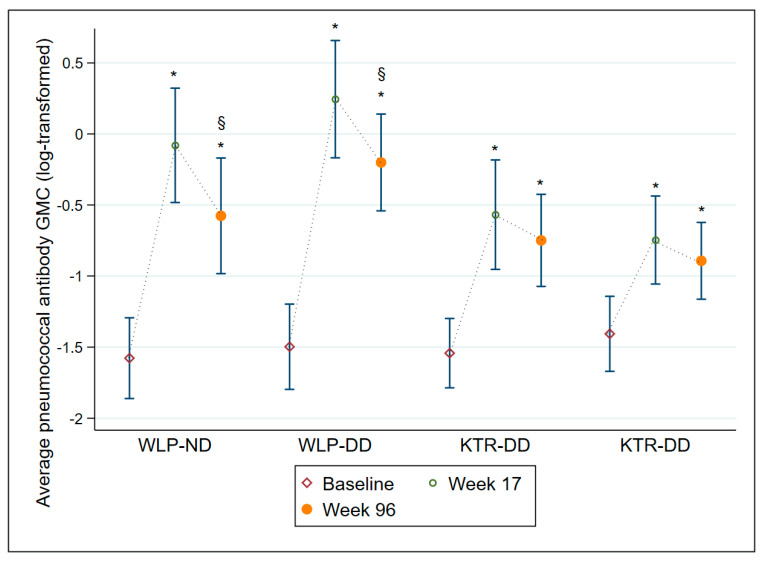
Average pneumococcal antibody geometric mean concentrations with 95% confidence intervals at baseline, week 17, and week 96 on a log-transformed scale for each of the 4 treatment groups. Abbreviations: WLP-ND, patients on the kidney transplant waiting list—normal dose; WLP-DD, patients on the kidney transplant waiting list—double dose; KTR-ND, kidney transplant recipient—normal dose; KTR-DD, kidney transplant recipient—double dose; GMC, Geometric mean concentration; CI, confidence interval. *: *p* ≤ 0.001, compared to baseline within group, §: *p* ≤ 0.001, compared to week 17 within group.

**Figure 4 vaccines-10-01091-f004:**
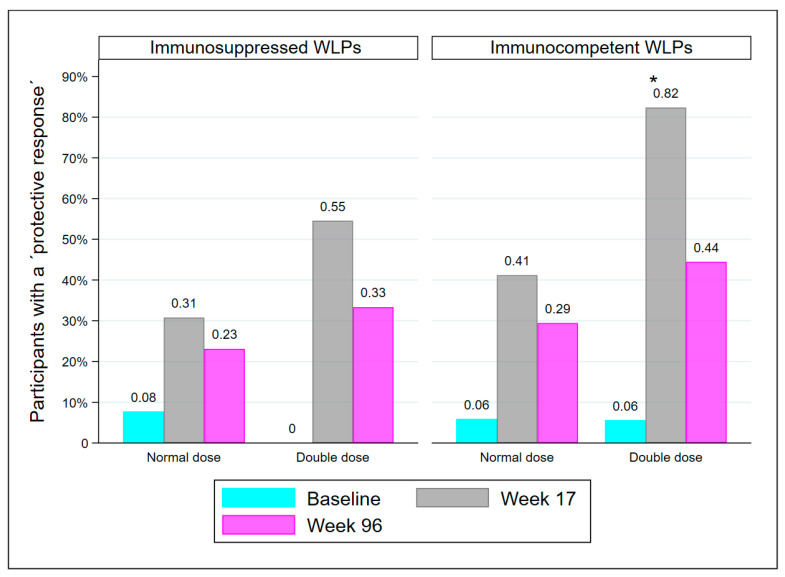
Immunocompetent WLPs and medical immunosuppressed WLPs who are protected at baseline, week 17, and week 96, divided into dose groups. Abbreviations: WLP, patients on the kidney transplant waiting list, *: *p* < 0.05, compared to immunocompetent WLP-ND at week 17.

**Table 1 vaccines-10-01091-t001:** Baseline demographics, immunosuppressive treatments, and kidney transplants performed during the trial.

	WLP-ND	WLP-DD	*p*-Value	KTR-ND	KTR-DD	*p*-Value
	N = 30	N = 30		N = 37	N = 33	
**Age, years, median (IQR)**	54.5 (44–59)	51 (39–62)	0.701	50 (41–62)	51 (43–60)	0.828
**Males, N (%)**	16 (53.3)	20 (66.7)	0.292	27 (72.9)	27 (81.8)	0.379
**Immunosuppressants, N (%)**						
Tacrolimus	1 (3.3)	2 (6.7)	1.000	36 (97.3)	32 (97)	1.000
Cyclosporine	1 (3.3)	0	1.000	0	1 (3)	0.471
Mycophenolic acid	2 (6.7)	5 (16.7)	0.424	36 (97.3)	33 (100)	1.000
Steroids	5 (16.7)	6 (2.0)	1.000	13 (35.1)	10 (33.3)	0.667
Various §	2 (6.7)	2 (6.7)	1.000	2 (5.4)	0	1.000
**Doses, median (IQR)**						
Mycophenolate mofetil g/day	NA	NA		1.5 (1.5–2)	1.5 (1.5–2)	0.914
Tacrolimus mg/day	NA	NA		3.5 (3–5)	4 (3–6)	0.178
Prednisolone mg/day	NA	NA		5 (5–7.5)	7.5 (5–10)	0.367
**Days since KTX, median (IQR)**	NA	NA		147 (83–271)	128 (71–288)	0.525
**KTXs, N (%)**			0.292			
Before week 17	7 (23.3)	3 (1.0)		NA	NA	
After week 17	11 (36.7)	10 (33.3)		NA	NA	
Not performed	12 (4.0)	17 (56.7)		NA	NA	

Abbreviations: WLP-ND, patients on the kidney transplant waiting list—normal dose; WLP-DD, patients on the kidney transplant waiting list—double dose; KTR-ND, kidney transplant recipients—normal dose; KTR-DD, kidney transplant recipients—double dose; IQR, interquartile ratio; KTX, kidney transplantation; NA, not applicable. §: Consists of Everolimus, Quinine, Azathioprine, or Secukinumab.

**Table 2 vaccines-10-01091-t002:** Participants protected at baseline, week 17, and week 96.

	WLP-ND	WLP-DD	*p*-Value	KTR-ND	KTR-DD	*p*-Value
**Visit**	N (%)	N (%)		N (%)	N (%)	
**Baseline**	2 (6.7)	1 (3.3)	1.000	0	1 (3.0)	0.471
**Week 17**	11 (36.7)	20 (71.4)	0.008	13 (35.1)	7 (21.2)	0.198
**Week 96**	8 (26.7)	12 (40) §	0.273	8 (21.6)	3 (9.1)	0.197

Abbreviations: WLP-ND, patients on the kidney transplant waiting list—normal dose; WLP-DD, patients on the kidney transplant waiting list—double dose; KTR-ND, kidney transplant recipient—normal dose; KTR-DD, kidney transplant recipient—double dose. §: *p* < 0.05, compared to week 17 within group.

**Table 3 vaccines-10-01091-t003:** Average pneumococcal antibody geometric mean concentrations with 95% confidence intervals at baseline, 17 weeks, and 96 weeks (mg/L).

	Baseline	Week 17	Week 96
**Group**	GMC (95% CI)	GMC (95% CI)	GMC (95% CI)
**WLP-ND**	0.21(0.16–0.27)	0.92 (0.62–1.38) *	0.56 (0.27–0.84) *§
**WLP-DD**	0.21 (0.16–0.28)	1.28 (0.85–1.93) *	0.77 (0.55–1.07) *§
**KTR-ND**	0.21 (0.17–0.27)	0.56 (0.39–0.83) *	0.47 (0.34–0.65) *
**KTR-DD**	0.24 (0.19–0.32)	0.47 (0.38–0.65) *	0.41 (0.31–0.54) *

Abbreviations: WLP-ND, patients on the kidney transplant waiting list—normal dose; WLP-DD, patients on the kidney transplant waiting list—double dose; KTR-ND, kidney transplant recipient—normal dose; KTR-DD, kidney transplant recipient—double dose; GMC, Geometric mean concentration; CI, confidence interval. *: *p* ≤ 0.001, compared to baseline within group. §: *p* ≤ 0.001, compared to week 17 within group.

## Data Availability

The data that support the findings of this study are available from the corresponding author upon reasonable request.

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
