# Peer review of "Durability of Antibody Response after Primary Pneumococcal Double-Dose Prime-Boost Vaccination in Adult Kidney Transplant Recipients and Candidates: 18-Month Follow-Up in a Non-Blinded, Randomised Clinical Trial"

_vaccines, 2022, doi:10.3390/vaccines10071091_

Round 1

Reviewer 1 Report

This is a follow up study on the antibody responses generated by patients either on dialysis or who are post renal transplantation to two different vaccination regimens to pneumococcus.

It is already known that the antibody responses can decay with time in these patient populations and that measuring other markers such as T cell counts does not correlate with vaccine efficacy. Plus in some countries a double dose of pneumococcal vaccination is already recommended. Hence can you provide some background context for your region-ie what were the national recommendations for pneumococcal vaccination in vulnerable populations prior to now and has your research led to any alteration in the strategy? Plus no mention is made of how you would manage the renal transplant patients in the future who have not received two doses of the vaccine whilst on the waiting list

Of note the end of the first sentence of the conclusions in the abstract needs to be modified to clarify what you mean by participant protected declined

At one stage in the discussion you mention that " This may be influenced by KTRs in our study being relatively newly transplanted, and the WLPs having negligible co-morbidity, compared to patients on dialysis" However you do not provide enough data to support this. Do you mean the number of patients who were vaccinated within 3-6 months of being transplanted and/or were transplanted shortly after being vaccinated whilst on the waiting list? Plus what other co-morbidities might potentially be an issue?

Finally the limitations of this study need to be explored in more detail in the discussion-

1) That is the timing of the vaccinations in this particular study cohort. There is concern as to what is the optimal timing of pneumococcal vaccination post kidney transplantation-possibly not in the first 6 months and or whilst the recipients are on the peak MMF dosages

2) Plus the timing between dosages may not have been optimal, so what would need to happen in order to further understand how to optimize the dosing schedule?

3) That a longer follow up period is required in order to understand if vaccine efficacy is being maintained (ie that there are no subsequent pneumococcal infections in this particular cohort)

4) What in the way of further research is now required in order to understand as to when patients can be considered to be fully vaccinated against this pathogen and how will this be actually measured?

Author Response

Response to reviewer 1

We thank the reviewer for the response and suggestions, which improved our manuscript. We have tried to the best of our abilities to answer the questions and improve the quality of the work done.

We would also like to inform the reviewer that due to sentence similarities with the first article from the trial conducted, we have made sentence changes without new or added information. We are sorry for any inconvenience this makes for your work.

Kind regards

Lykke Larsen, on behalf of all authors.

Comments and Suggestions for Authors

This is a follow up study on the antibody responses generated by patients either on dialysis or who are post renal transplantation to two different vaccination regimens to pneumococcus.

It is already known that the antibody responses can decay with time in these patient populations and that measuring other markers such as T cell counts does not correlate with vaccine efficacy. Plus in some countries a double dose of pneumococcal vaccination is already recommended. Hence can you provide some background context for your region-ie what were the national recommendations for pneumococcal vaccination in vulnerable populations prior to now and has your research led to any alteration in the strategy? Plus no mention is made of how you would manage the renal transplant patients in the future who have not received two doses of the vaccine whilst on the waiting list

Reply: We have added a section in the Introduction on the national recommendation and practice. No alterations to the vaccine strategy has been made yet.

Of note the end of the first sentence of the conclusions in the abstract needs to be modified to clarify what you mean by participant protected declined

Reply. The sentence has been changed.

At one stage in the discussion you mention that " This may be influenced by KTRs in our study being relatively newly transplanted, and the WLPs having negligible co-morbidity, compared to patients on dialysis" However you do not provide enough data to support this. Do you mean the number of patients who were vaccinated within 3-6 months of being transplanted and/or were transplanted shortly after being vaccinated whilst on the waiting list? Plus what other co-morbidities might potentially be an issue?

Reply: We have edited the sentence and removed the WLP part, as it is true we might only speculate that WLPs are younger and healthier than patients on dialysis are in general.

Finally the limitations of this study need to be explored in more detail in the discussion-

1) That is the timing of the vaccinations in this particular study cohort. There is concern as to what is the optimal timing of pneumococcal vaccination post kidney transplantation-possibly not in the first 6 months and or whilst the recipients are on the peak MMF dosages

Reply: We agree that this may be a contributing factor and have added a section in the discussion about timing of vaccination.

2) Plus the timing between dosages may not have been optimal, so what would need to happen in order to further understand how to optimize the dosing schedule?

Reply: We have added a section in the discussion about time-interval between the vaccines.

3) That a longer follow up period is required in order to understand if vaccine efficacy is being maintained (ie that there are no subsequent pneumococcal infections in this particular cohort)

Reply: We have added a section in the discussion

4) What in the way of further research is now required in order to understand as to when patients can be considered to be fully vaccinated against this pathogen and how will this be actually measured?

Reply: We have added a section in the discussion addressing vaccine efficacy studies in our populations.

Reviewer 2 Report

Thank you for the opportunity to review the manuscript by Lykke Larsen et al entitled “Durability of antibody response after primary pneumococcal double dose prime-boost vaccination in adult kidney transplant recipients and candidates: 18 months follow-up in a non-blinded, randomized clinical trial.” 

The authors have evaluated long term effects of a double or single dose of pneumococcal vaccines and conclude that wait listed patients (WLP) receiving a double dose had better short time (17 weeks) effect than those receiving a single dose, but that the effect declined in both groups and after 96 weeks there were no statistical significant differences between the two groups of WLP. Kidney transplant recipients (KTR) on the other hand showed no effect of the double dose neither after 17 nor after 96 weeks. 

The manuscript is well-written and addresses a topic of major interest and high relevance for clinicians treating immunosuppressed patients, especially in accordance with the knowledge that have been gained about vaccine response in KTRs during the covid19 pandemic.

I have some remarks that the authors should address.

Major:

1.     The authors have defined an antibody level of > 1 mg/L as protective response (PR) at 96 weeks. This is according to the national Danish practice. What is the basis for the national practice? At least a reference to this practice should be added although the choice of cut-off for PR is discussed later in the manuscript where the authors also state that there is no consensus on protective levels for adult SOT or adults in general.

2.     Using the PR cut-off value of 1 mg/L, only 16% of KTR and 40% of WLP had antibody level above PR at 96 weeks. In contrast to this, the authors conclude that vaccination provides a durable antibody response in both KTR and WLP. The authors should discuss whether the low proportion of KTR having PR at 96 weeks indicates that KTR for example should receive a booster dose after some time (12 months or when antibody level falls below PR) to maintain the protection against pneumococcal infection. A future booster dose should of course be evaluated in future studies.

3.     The authors argue that they will recommend double dosage for WLP even though there was no significant difference in response between ND and DD after 96 weeks. What are the arguments for this conclusion? The difference in proportions is relatively large (40% vs 27%) which can indicate that the lack of statistical significance is a consequence of the underpowering of the study. It is also somewhat unexpected that KTR seem to have less response from DD than ND both after 17 and 96 weeks.

4.     The main limitation of the study is, as mentioned by the authors, that it is underpowered. Consequently, one should not draw firm conclusions, but the novel findings should be explored in further studies. This should be stated more clearly in the manuscript.

Minor:

1.     Table 1: Was tacrolimus the only CNI used by the patients in the study or did some patients receive CsA? If yes, I would like to know the proportions of patients using tac and CsA

2.     Table 1: What was the tacrolimus concentration? Concerning degree of immunosuppression, tacrolimus concentration will be of more value than the dosage.

3.     Typo error in Table 1: 10 patients in the KTR-DD received steroids, this is 30.3% not 3.3 as reported in the table

4.     Conclusions: The authors conclude that the numbers of participants with an estimated PR dropped, I suggest the change to state that the proportion dropped.

Author Response

Response to reviewer 2

We thank the reviewer for the response and suggestions, which improved our manuscript. We have tried to the best of our abilities to answer the questions and improve the quality of the work done.

We would also like to inform the reviewer that due to sentence similarities with the first article from the trial conducted, we have made sentence changes without new or added information. We are sorry for any inconvenience this makes for your work.

Kind regards

Lykke Larsen, on behalf of all authors.

Comments and Suggestions for Authors

Major:

  1. The authors have defined an antibody level of > 1 mg/L as protective response (PR) at 96 weeks. This is according to the national Danish practice. What is the basis for the national practice? At least a reference to this practice should be added although the choice of cut-off for PR is discussed later in the manuscript where the authors also state that there is no consensus on protective levels for adult SOT or adults in general.

Reply: A reference has been added. Only one lab in Denmark (Statens Serum Institute) runs the test for serotype-specific IgG antibodies. The level is one that they estimated appropriated for Luminex compared to the ELISA test and the literature on the subject. However, no really god reference exist.

  1. Using the PR cut-off value of 1 mg/L, only 16% of KTR and 40% of WLP had antibody level above PR at 96 weeks. In contrast to this, the authors conclude that vaccination provides a durable antibody response in both KTR and WLP. The authors should discuss whether the low proportion of KTR having PR at 96 weeks indicates that KTR for example should receive a booster dose after some time (12 months or when antibody level falls below PR) to maintain the protection against pneumococcal infection. A future booster dose should of course be evaluated in future studies.

Reply: considerations regarding a booster doses have been added to the discussion.

  1. The authors argue that they will recommend double dosage for WLP even though there was no significant difference in response between ND and DD after 96 weeks. What are the arguments for this conclusion? The difference in proportions is relatively large (40% vs 27%) which can indicate that the lack of statistical significance is a consequence of the underpowering of the study. It is also somewhat unexpected that KTR seem to have less response from DD than ND both after 17 and 96 weeks.

Reply: The conclusion have been rewritten so that the lack of statistical significance is taken into account. Limitations have been added a sentence on “firm conclusions cannot be draw,” as the study is underpowered.

  1. The main limitation of the study is, as mentioned by the authors, that it is underpowered. Consequently, one should not draw firm conclusions, but the novel findings should be explored in further studies. This should be stated more clearly in the manuscript.

Reply: This has been added to the limitations section.

Minor:

  1. Table 1: Was tacrolimus the only CNI used by the patients in the study or did some patients receive CsA? If yes, I would like to know the proportions of patients using tac and CsA

Reply: We have added cyclosporine to Table 1.

  1. Table 1: What was the tacrolimus concentration? Concerning degree of immunosuppression, tacrolimus concentration will be of more value than the dosage.

Reply: Unfortunately, we did not obtain this measurement on the patients, but obtained their medication and changes of these from their electronic medication card. We are therefore not in a position to provide this information.

  1. Typo error in Table 1: 10 patients in the KTR-DD received steroids, this is 30.3% not 3.3 as reported in the table

Reply: Thank you for this observation. It has been corrected.

  1. Conclusions: The authors conclude that the numbers of participants with an estimated PR dropped, I suggest the change to state that the proportion dropped.

Reply: Proportion has been added instead in the abstract and the conclusion